# Economic burden of chronic pain in Alberta, Canada

Nguyen Xuan Thanh[1,2]*, Robert L. Tanguay[3], Kiran J. Pohar Manhas[1,3], Ania Kania-Richmond[1,3], Sherri Kashuba[1], Tracey Geyer[4], John X. Pereira[3], Tracy Wasylak[1,5]

1 Strategic Clinical Networks™, Alberta Health Services, Edmonton, Calgary, and Red Deer, Alberta, Canada, 2 School of Public Health, University of Alberta, Edmonton, Alberta, Canada, 3 Cumming School of Medicine, University of Calgary, Calgary, Alberta, Canada, 4 Planning and Performance, Alberta Health Services, Edmonton, Alberta, Canada, 5 Faculty of Nursing, University of Calgary, Calgary, Alberta, Canada

* thanh.nguyen3@ahs.ca

**Data Availability Statement:** The minimal data are included in the Supporting Information files. Furthermore, the Public Use Microdata File of the CCHS 2013/2014, which was used for estimating the prevalence of chronic pain, the utilization of

## Abstract

### Background

Although chronic pain (CP) is common, little is known about its economic burden in Alberta, Canada.

### Aims

To estimate incremental (as compared to the general population or people without CP) societal (healthcare and lost productivity) costs of CP in Alberta.

### Methods

We applied the prevalence estimated from the Canadian Community Health Survey data to the population retrieved from the Statistics Canada to estimate the number of people with CP in Alberta in 2019. We analyzed the Alberta Health administrative databases to estimate the healthcare costs of person with CP. Finally, we multiplied the number of people with the cost per person.

### Results

The prevalence of any CP was 20.1% and of activity-preventing CP was 14.5% among people aged > = 12 years. Incremental cost per person with CP per year was CA$2,217 for healthcare services (among people aged > = 12 years) and CA$8,412 for productivity losses (among people aged 18–64 years). Of the healthcare cost, prescription drugs accounted for the largest share (32.8%), followed by inpatient services (31.0%), outpatient services (13.1%), physician services (9.8%), other services (7.4%), and diagnostic imaging (5.8%). Provincially, total incremental cost of CP ranges from CA$1.2 to 1.7 billion for healthcare services (6% to 8% of total provincial health expenditure); and CA$3.4 to 4.7 billion for productivity losses. Considering costs for long-term care services, the total societal cost of CP in Alberta was CA$6.3 to 8.3 billion per year, reflecting 2.0% to 2.7% of Alberta's GDP.

other health services, and the employment-related information, can be requested for free at Products and services order form (statcan.gc.ca).

**Funding:** The author(s) received no specific funding for this work.

**Competing interests:** The authors have declared that no competing interests exist.

## Conclusions

Interventions improving CP prevention and management to reduce this substantial economic burden are urgently needed.

## Introduction

Chronic pain (CP) is defined as pain that persists or recurs for longer than 3 months [1]. It has been recognized as a disease in its own right and can be primary when associated with emotional distress and functional disability, or secondary when following other conditions (post-surgery or post-trauma, etc.). There are multiple factors that influence the development and experience of CP, including biomedical, psychological, social, environmental, emotional, and spiritual [2].

CP is one of the leading causes of disability and morbidity in Canada and globally. It is estimated that 19 to 22% of Canadians aged 18 or older experience CP [3, 4]. CP carries both human and economic costs for patients, families, communities, and society. Previous studies concluded that, in addition to direct healthcare costs, pain undermines one's ability to participate fully in relationships, schools, workplaces, and communities. According to Gaskin & Richard [5], in the United States, the total annual cost of CP is estimated from US$560 to 635 billion, including the incremental healthcare cost of US$261 to 300 billion and the lost productivity cost of US$297 to 336 billion (US$1 = CA$1.28). The estimate in Australia is AU$73.2 billion in 2018 (AU$1 = CA$0.94), of which lost productivity accounts for 66% (AU$48.3 billion), health system costs for 17% (AU$12.2 billion) and other financial costs (informal care, aids and modifications and deadweight losses) for 17% (AU$12.7 billion) [6]. In Canada, while the indirect costs are not readily available, the annual incremental healthcare cost to manage CP is estimated at CA$7.2 billion or CA$1,742 per patient in 2014 [7].

While the prevalence of CP in Alberta (a western province of Canada with a population of approximately 4.4 million [8] and a publicly funded healthcare system) was estimated more than a decade ago (between Oct. 2007 and Oct. 2008) at 20.6% [3], the economic burden of CP is still unknown. Recently, a multi-stakeholder group from across Alberta developed the Alberta Pain Strategy 2019–2024 with a vision of "achieving excellence in pain management across the lifespan for all Albertans" [9]. The strategy was based on local successes supplemented by existing national and international evidence supporting the spread and scale of innovative models of care to improve access to appropriate care delivery, education, and outcomes for all Albertans. While the strategy outlines specific actions for several areas of pain, the operational leadership within Alberta's healthcare system prioritized CP as an important area of initial focus aiming to support people living with CP, their families/caregivers, and providers to optimize management of CP and its' impacts on individuals' function and quality of life, as well as on societal economic burden.

In this study, we aimed to estimate societal (healthcare and lost productivity) costs of CP in Alberta to fill the knowledge gap and to establish a baseline for future evaluations of the implementation of interventions within the Alberta Pain Strategy.

## Materials and methods

The economic burden (EB) of CP was estimated in three steps. First, we estimated the number of people with CP (N). Second, we estimated the cost per person with CP (C). Finally, the

economic burden was estimated by multiplying the number of people with CP with the cost per person (EB = N x C).

## Methods for estimating the number of people with CP

Since the International Classification of Disease (ICD) codes of CP (ICD-9: 338 and ICD-10: G89) were not used in the Alberta Health administrative databases, we used the Canadian Community Health Survey (CCHS) data to estimate the prevalence of CP in Alberta. The CCHS is national cross-sectional survey of Canadians aged 12 years and above with a sample of appropriately 120,000 people, including about 12,000 Albertans. The survey has been carried out every two years asking questions related to health status, healthcare utilization, determinants of health, etc. Some of the contents are changed by survey cycle and province [10].

As the pain and discomfort questions were unavailable in the most recent cycles of CCHS, we used CCHS 2013–2014 where the pain and discomfort questions were a core content that all provinces, including Alberta, asked surveyed respondents. In the CCHS microdata for public use, respondents were categorized into 5 groups: "1-no pain or discomfort", "2-pain prevents no activities", "3-pain prevents a few activities", "4-pain prevents some activities", and "5-pain prevents most activities".[9] We used the data of Alberta only and calculated the prevalence (weighed by the sampling weights) separately between any CP (groups 2–5) and activity-preventing CP (groups 3–5). To estimate the number of people with the disease in Alberta, we multiplied this prevalence with Alberta's population (obtained from Statistics Canada) [8]. To better understand the distribution of CP, we analyzed the data by age group (12 to 17 years; 18 to 64 years, and 65 years or older).

## Methods for estimating the cost per person with CP

**Healthcare cost.** We estimated 1-year healthcare costs of CP, including the following health services or cost components: inpatient, outpatient, physician, diagnostic imaging, prescription drugs, and others (such as nurse, physiotherapist, chiropractor, psychologist, social worker, and occupational therapist). We estimated an incremental cost, which is the additional cost of CP compared to that of the general population or people without CP.

The utilizations of nurse, physiotherapist, chiropractor, psychologist, social worker, and occupational therapist services were estimated from the CCHS data. In this survey, the respondents were asked about their utilizations of these services in the past 12 months. We calculated differences in the utilizations (i.e. numbers of visits) of these services between people with and without CP. We then multiplied these differences with respective unit costs (S1 Table in S1 File) to obtain the incremental cost for these services per year.

Costs for inpatient services, outpatient services, physician services, prescription drugs, and diagnostic imaging (DI) services were estimated from the Alberta Health administrative databases [11]. Because the ICD codes of CP were not used, health services utilization (HSU) costs of CP were estimated by multiplying the costs of 12 CP-related conditions (that can be identified by ICD codes–S2 Table in S1 File) with respective weights (proportion of CP due to condition) as reported by Pain Australia.[6] These conditions (weights) included injury (0.380), musculoskeletal (0.241), cancer (0.016), mental health/behavioural (0.011), gastrointestinal (0.01), neurological (0.007), circulatory (0.007), infection (0.006), genitourinary (0.006), endocrine/hormonal (0.002), respiratory (0.002), and unknown (0.314) [6, 12, 13]. Of note, the sum of weights is 1 (or 100%). For simplification, the weight of unknown was proportionately distributed to the 11 identifiable conditions for the HSU cost calculation (S3 Table in S1 File).

For each condition, costs for inpatient and outpatient services were based on the Canadian Institute for Health Information (CIHI) Case-Mix Group Plus (CMG+) methodology, which

included both medical and non-medical (e.g. support and administrative departments, such as information systems, housekeeping, finance, etc.) costs [14]. The cost for each CMG+ or Comprehensive Ambulatory Classification System (CACS) group was retrieved from the Alberta Health Interactive Health Data Application [15]. The cost for physician services was defined as paid amounts available in the physician claims to the Healthcare Insurance. Costs for prescription drugs were based on prices per unit listed in the Alberta Drug Benefit List [16]. Costs for diagnostic imaging, such as cost per CT, MRI, or X-Ray, were retrieved from Alberta Health Services finance and the published literature. Of note, transfer funds from an insurer (e.g. auto insurance company) to Alberta Health were not considered a cost and therefore not included in this study. The medical aids from the Worker's Compensation Board (a workers' insurer) to workers/patients regarding costs for nurses, physiotherapy, chiropractic, acupuncture, psychology, etc. were already included in the costs for these services estimated from the CCHS data mentioned earlier.

The incremental cost for HSU was then calculated by comparing the cost of HSU per person with CP estimated as mentioned above with the HSU cost per average person in the general population that was reported previously [17].

**Lost productivity cost.** We used a modeling technique to estimate the incremental cost, which was defined as the difference in cost of productivity losses between people with and without chronic pain. We conservatively included people at working ages (18 to 64 years old) and excluded those with chronic pain aged 12 to 17 years and aged 65 years or older from this analysis.

Productivity losses included four components: unable to work permanently, unemployment, absenteeism, and presenteeism. Premature death was not considered, as previous studies suggest a minimal impact [6]. We applied a human capital approach involving multiplying the number of missed working year, day, or hour per person with the average wage per person per year, day, or hour. We estimated the cost for the year of 2019, when the average wage was $55,900 per person, per year [18]. We divided this wage by 251 working days per year (www.workingdays.ca) to estimate the average wage per day. The wage per hour was estimated by dividing the wage per day by 8 hours.

The model structure consists of, and compares, two arms: CP vs. No CP (Fig 1). A person with or without CP can be either able or unable to work. An able-to-work person can be either

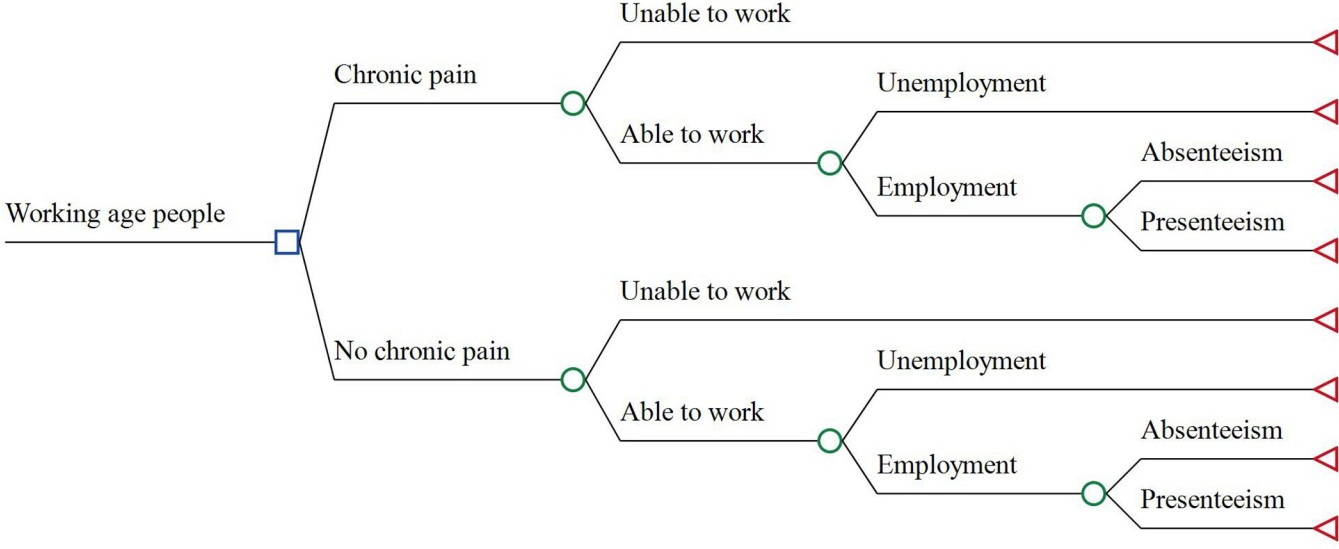

**Fig 1. Model structure to estimate the incremental cost of lost productivity.**

employed or unemployed. An employee would miss an average number of working days (absenteeism) and an average number of working hours (presenteeism). Although the structure is identical, the two arms are different in terms of probabilities or percentages of the unable-to-work and/or unemployed, as well as in terms of the number of missed working days and/or hours. It is expected that people with chronic pain would have a higher percentage of those unable to work, a higher percentage of unemployment, and a higher number of missed working days and hours, thereby a higher cost of productivity losses, in comparison with their counterparts (i.e., the incremental cost is greater than zero).

Model inputs (S4 Table in S1 File) were estimated using the microdata of CCHS 2013/2014 given the availability of chronic pain, labor participation, and lost productivity questions in this survey cycle [10]. We estimated the percentages of respondents who were permanently unable to work among the CP and no CP groups (CCHS variable use: LBSDWSS–working status last week). The percentages of unemployment among those who were able to work for CP and no CP groups were estimated using the variable "LOP_015" regarding employment status last 3 months. The average numbers of missed working days were estimated using the variable "LOPG040" where respondents were grouped in to 1 to 30 and 31+. Conservatively, we used 31 days for all people reporting 31 days or more. All respondents who reported "no absent" for variable "LOP_030" regarding missed work due to chronic condition, were considered 0 day of absenteeism.

For presenteeism, as it is not available in the CCHS data, we retrieved the average number of missed working hours among people with and without CP from Kawai et al. [19]. We multiplied the 3-month estimates (e.g. the missed working days per 3 months in the CCHS data) with 4 and the weekly estimates (e.g. the missed working hours per week in the study by Kawai et al. [19]) with 50 (the number of working weeks per year), to obtain an annual estimate. Also, we assumed that those who reported "permanently unable to work" last week and who reported "unemployment" last 3 months, were unable to work or unemployed for the whole year.

We performed both deterministic and probabilistic sensitivity analyses for the uncertainty of all of these parameters. In the deterministic sensitivity analysis, we performed a one-way analysis (i.e. one variable varied at a time) for all variables allowing each variable to vary from the lower to the upper value of its 95% confidence interval. For the average wage, we assumed it varied by ±30%. In the probabilistic sensitivity analysis (all variables varied simultaneously), we ran 100,000 trials/samples to estimate a 95% confidence interval of the incremental cost of productivity losses, assuming a beta distribution for probabilities, a gamma distribution for costs, and a Poisson distribution for numbers of missed working days and hours.

## Methods for calculating the total cost/economic burden of CP for Alberta

The total cost was calculated by multiplying the number of people with CP (section 1) with the incremental cost per person with CP (section 2). We reported a range of the total cost, where the lower value corresponded to the number of people with activity-preventing CP and the higher value corresponded to the number of people with any CP. In addition to the absolute dollar amount, we relatively calculated the total healthcare cost as a percentage of the total healthcare expenditure in the province and the total societal cost as a percentage of the provincial gross domestic product (GDP).

All costs were inflated to 2020 Canadian dollars (CA$) using the Bank of Canada Inflation Calculator [20]. Stata SE version 16.0 (www.stata.com), TreeAge Pro 2019 R2.0 (www.treeage.com) and Microsoft Excel (www.microsoft.com) were used for data analyses.

As this study was a secondary analysis of previously collected data by Statistics Canada and Alberta Health, the Research Ethics Board (REB) at the University of Alberta indicated that

obtaining additional informed consent was not required. It was ethically approved by the REB (File # Pro00110848) on May 25, 2021.

## Results

There were totally 12,072 people aged 12 years or older in Alberta participating in the CCHS survey. Males accounted for 46.2% and females for 53.8%. People aged 12 to 17 years accounted for 8.2%, 18 to 64 accounted for 66.3%, people aged 65 years or older for 25.4%.

Weighted prevalence of CP by age group is shown in Table 1. For all groups, prevalence of any CP was 20.1% and of activity-preventing CP was 14.5%. Our results showed that increasing age was associated with higher prevalence. The highest prevalence (29.7% for any CP and 21.1% for activity-preventing CP) was found in the oldest age group (65 years or older), followed by people aged 18 to 64 years (19.8% for any CP and 14.5% for activity-preventing CP). The lowest prevalence was found in the youngest group of 12 to 17 years (8.3% for any CP and 3.7% for activity-preventing CP).

Applying the prevalence rates to the population by age group (S5 Table in S1 File [8]), the number of people with CP was estimated (Table 2). There were totally 755,957 people with CP in Alberta, of which 542,447 (72%) with activity-preventing CP. About 74% of people with CP were aged 18 to 64 years, followed by people aged 65 years or older (~23%), and aged 12 to 17 years (~3%).

Average HSU cost per person with injury per year was estimated at CA$3,391. This amount for cancer was CA$9,401; musculoskeletal CA$5,237; mental health CA$13,057; gastrointestinal CA$6,196; neurological CA$8,772; infection CA$4,259; circulatory CA$3,991; genitourinary CA$4,466; endocrine/hormonal CA$7,052; and respiratory CA$4,268, respectively. Multiplying these costs with respective weights (proportion of CP due to related conditions) and then summing them up, the annual HSU cost per person with CP was estimated at CA$4,464 (S3 Table in S1 File). From this, we subtracted the HSU cost per average person in the general population (CA$2,442 [17]) to estimate the incremental HSU cost per person with CP at CA$2,022. Adding the incremental cost of other services (e.g. nurse, physiotherapist, chiropractor, psychologist, social worker, and occupational therapist) which was estimated from the CCHS data at CA$195, the total incremental healthcare cost per person with CP per year was estimated at CA$2,217 (Table 3). Of this cost, prescription drugs accounted for the largest

**Table 1. Prevalence of chronic pain in Alberta by age group.**

| Age group | Activity-preventing chronic pain | Any chronic pain |
|---|---|---|
| 12 to 17 years | 3.70% | 8.30% |
| 18 to 64 years | 14.50% | 19.80% |
| 65+ years | 21.10% | 29.70% |
| All ages | 14.50% | 20.10% |

**Table 2. Number of people with chronic pain in Alberta by age group.**

| Age group | Activity-preventing chronic pain | Any chronic pain |
|---|---|---|
| 12 to 17 years | 11,460 (2.1%) | 25,927 (3.4%) |
| 18 to 64 years | 408,576 (75.3%) | 557,958 (73.8%) |
| 65+ years | 122,410 (22.6%) | 172,072 (22.8%) |
| All ages | 542,446 (100%) | 755,957 (100%) |

**Table 3. Incremental cost ($) per person and total cost (billion $) of chronic pain per year for Alberta.**

| Cost category | | Activity-preventing chronic pain | Any chronic pain |
|---|---|---|---|
| Healthcare | Cost per person ($) | $2,217.27 | |
| | Total (B$) | 1.20 | 1.68 |
| Lost productivity | Cost per person ($) | 8,412.00 | |
| | Total (B$) | 3.44 | 4.69 |
| Total societal costs (B$) | | 4.64 | 6.37 |

share (33%), followed by inpatient service (30%), outpatient service (13%), physician service (10%), other services (7%), and diagnostic imaging (6%) (S1 Fig).

The incremental cost of lost productivity per person with CP was estimated at CA$8,412 (Table 3). This was the difference between costs of lost productivity per person with and without CP which were estimated at CA$16,960 and CA$8,548, respectively (S6 Table in S1 File). Sensitivity analyses showed that the incremental cost varied between CA$5,327 and CA$11,498.

Multiplying the number of people with CP (Table 2) with the incremental cost per person (Table 3), total annual incremental cost of CP for Alberta was estimated from CA$1.2 to 1.7 billion for healthcare services and from CA$3.4 to 4.7 billion for productivity losses (Table 3). According to the Government of Alberta [21], total health expenditure was CA$20.6 billion in 2019. Therefore, the total healthcare cost of CP as a percentage of the total health expenditure was estimated from 5.9% to 8.3%.

## Discussion

We analyzed the Canadian Community Health Survey and Alberta Health administrative databases, and used a decision analytic model with a human capital approach to estimate prevalence and incremental healthcare and lost productivity costs of CP in the province of Alberta. The results show that the prevalence of any CP is 20.1% and of activity-preventing CP is 14.5%, among people aged 12 years or older. The incremental cost per person with CP per year is estimated at CA$2,217 for healthcare services and CA$8,412 for productivity losses. Provincially, the total incremental cost of CP is CA$1.2 to 1.7 billion for healthcare services and CA$3.4 to 4.7 billion for productivity losses. Together, the total healthcare and lost productivity cost of CP in Alberta is estimated from CA$4.6 to 6.4 billion per year, of which healthcare accounts for 26% and lost productivity 74%. That is the lost productivity cost is almost 3 time higher than the healthcare cost. This aligns with results of the national estimates from Australia where the lost productivity cost is reported to be approximately 4 times higher than the healthcare cost [6].

Regarding healthcare services, the total cost accounts for 6 to 8% of the total provincial health expenditure in 2019. This percentage is even higher than the percentage of total spending for all cancers (4.8%) or diseases of respiratory (5.9%) or musculoskeletal systems (6%) in Canada [22].

Our findings are comparable to other studies in Canadian or similar contexts. For example, Schopflocher and colleagues performed a telephone survey to a representative sample of adults aged 18 years or older from across Canada between 2007 and 2008 and found that the prevalence of CP in the province of Alberta was 20.6% [3]. The authors also found that CP was more prevalent among older ages in comparison with younger, which is consistent with our findings. In terms of cost, Hogan and colleagues linked the CCHS data between 2000 and 2011 to the health administrative data in Ontario to estimate the incremental healthcare cost per

person with CP per year at CA$1,742 in 2014 [7] (~CA$1,936 in 2020). This is slightly lower but comparable to our findings because we included costs for other services (e.g. nurse, physiotherapist, chiropractor, psychologist, social worker, and occupational therapist), while the Ontario study did not. The total incremental healthcare cost of CP as a percentage of the total health expenditure is higher in our study compared to the Ontario study (6 to 8% vs. 5%). This can be explained by the fact that both the prevalence and the healthcare cost per person are higher in Alberta compared to Ontario or Canadian average. In Australia, it is estimated that the total health system cost of CP in 2018 was AU$12.2 billion [6], equating to more than 6% of the total health expenditure of that country (AU$195.7 billion [23]). This is comparable to our findings.

According to Guliani et al [24], among long-term care (LTC) residents aged 65+, people with clinically significant pain (defined as daily pain or moderate to severe nondaily pain) accounts for 35.9%, including 24.2% people with CP without comorbidities. Given the Alberta Government's expenditure on LTC services of CA$4 billion per year [25], we would expect approximately CA$1 billion is spent on LTC relating to CP. Additionally, compensation by the Worker's Compensation Board regarding re-employment and pension were estimated at CA $0.06 billion in 2019 [26]. Adding these to healthcare and lost productivity costs, the total cost of CP is CA$5.7 to 7.5 billion per year. Furthermore, given a unit cost of CA$1,400 (converted from $1,390 in 2018 Australian dollars [6]), the cost for lost productivity of informal caregivers (i.e., family members) for people with CP in Alberta would be estimated from CA$0.6 to 0.8 billion. In total, the cost of CP in Alberta is an estimated CA$6.3 to 8.3 billion per year. This reflects 2.0% to 2.7% of Alberta's GDP, which was CA$307.5 billion in 2020 [27] (Table 4).

Although slightly lower than that of Australia or European countries where the cost of CP is reported to be between 3% and 10% as a proportion of GDP [6, 28, 29], this is a significant economic burden. These variations can be explained by use of different cost components and costing methods between studies. For example, the study by Painaustralia [6] included costs for aids and modifications and deadweight losses while ours did not. Regarding costing methods in the main analyses, the study by Painaustralia used a top-down approach (money allocated to healthcare or social services) while our study used a bottom-up approach (cost per patient). Furthermore, we only estimated an incremental cost which may also be a reason why our estimate is lower than that of those who used a gross costing approach.

There are limitations to this study. First, as currently there are no ICD-9 or 10 codes for CP used in the Alberta Health admin databases, HSU costs of CP cannot be estimated directly, but through CP-related conditions and the weights–percentages of CP due to these conditions.

**Table 4. Total societal cost of chronic pain for Alberta.**

| Cost component | Low Billion $ (%) | High Billion $ (%) |
|---|---|---|
| *Direct cost*: | | |
| Healthcare services | 1.2 (19.2%) | 1.7 (20.6%) |
| Long-term care services | 1.0 (16.0%) | 1.0 (12.1%) |
| WCB compensations | 0.06 (1%) | 0.06 (0.7%) |
| *Indirect cost (Productivity losses)*: | | |
| People with chronic pain | 3.4 (54.3%) | 4.7 (56.9%) |
| Informal caregivers | 0.6 (9.6%) | 0.8 (9.7%) |
| **Total** | **6.3 (100%)** | **8.3 (100%)** |
| *Total as a percentage of GDP** | *2.0%* | *2.7%* |

*Based on GDP = $307.5 billion.[27]

Also, the prevalence of CP is subjective and not specific to pain lasting over 3 months as it is based on the CCHS data. In the future, when ICD-11 codes of CP are widely used, another study using the admin databases will be desirable. Second, as the utilization of other services was retrieved from the CCHS data, where respondents are asked to recall the number of visits that they made in the past 12 months, there may be some impact of recall bias. However, this is likely small for the total cost because the component of other services only accounts for a small portion (7%) of the total cost. Third, the lost productivity of people with chronic pain aged less than 18 or more than 64 years is not included. Given many people start working before the age of 18 and retire after the age of 65 [30], our estimate is conservative and the real total cost of lost productivity due to CP in Alberta is likely higher. Finally, in the CCHS 2013/2014, the labour participant section was done in Newfoundland and Labrador only and the rate could be different from that of Alberta. Furthermore, the numbers of missed working days or hours, the 'unable to work' and unemployment rates were extrapolated from a 3-month or 1-week period to 1 year. These factors would possibly make the incremental cost of lost productivity under- or over-estimated. However, the sensitivity analyses minimized these biases.

In conclusion, this study demonstrates that CP imposes a substantial economic burden on individuals, Alberta's healthcare system and economy. Interventions improving CP prevention and management to reduce the burden while improving the well-being of Albertans, are urgently needed. Having a baseline prevalence of CP and economic cost per patient will enable better assessment of the return on investment of CP interventions and provide concrete evidence to inform health resource allocation. It always enables comparisons across jurisdictions to better understand impact of healthcare services.

## Supporting information

**S1 File.**
(DOCX)

**S1 Fig. Distribution of healthcare costs for chronic pain in Alberta.**
(TIF)

## Author Contributions

**Conceptualization:** Nguyen Xuan Thanh, Robert L. Tanguay, Tracy Wasylak.

**Data curation:** Nguyen Xuan Thanh, Kiran J. Pohar Manhas, Ania Kania-Richmond, Sherri Kashuba.

**Formal analysis:** Nguyen Xuan Thanh, John X. Pereira.

**Funding acquisition:** Nguyen Xuan Thanh, Ania Kania-Richmond, Tracy Wasylak.

**Investigation:** Nguyen Xuan Thanh.

**Methodology:** Nguyen Xuan Thanh, Robert L. Tanguay, Sherri Kashuba, Tracey Geyer, John X. Pereira, Tracy Wasylak.

**Project administration:** Nguyen Xuan Thanh, Tracy Wasylak.

**Software:** Tracy Wasylak.

**Supervision:** Tracy Wasylak.

**Validation:** Kiran J. Pohar Manhas, Ania Kania-Richmond, Tracey Geyer, John X. Pereira.

**Writing – original draft:** Nguyen Xuan Thanh.

**Writing – review & editing:** Nguyen Xuan Thanh, Robert L. Tanguay, Kiran J. Pohar Manhas, Ania Kania-Richmond, Sherri Kashuba, Tracey Geyer, John X. Pereira, Tracy Wasylak.

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
