## [Decision Letter · Decision Letter 0]

9 Jun 2022

PONE-D-22-11788Economic Burden of Chronic Pain in Alberta, CanadaPLOS ONE

Dear Dr. Thanh,

Thank you for submitting your manuscript to PLOS ONE. After careful consideration, we feel that it has merit but does not fully meet PLOS ONE’s publication criteria as it currently stands. Therefore, we invite you to submit a revised version of the manuscript that addresses the points raised during the review process.

We look forward to receiving your revised manuscript.

Kind regards,

Vijay S. Gc, PhD

Academic Editor

PLOS ONE

Journal Requirements:

[The Bone & Joint Health Strategic Clinical NetworkTM of Alberta Health Services covered the publication fees.]

 [The author(s) received no specific funding for this work.]

Reviewers' comments:

Reviewer's Responses to Questions

**Comments to the Author**

1. Is the manuscript technically sound, and do the data support the conclusions?

Reviewer #1: Yes

Reviewer #2: Partly

2. Has the statistical analysis been performed appropriately and rigorously? 

Reviewer #1: Yes

Reviewer #2: Yes

3. Have the authors made all data underlying the findings in their manuscript fully available?

Reviewer #1: Yes

Reviewer #2: Yes

4. Is the manuscript presented in an intelligible fashion and written in standard English?

Reviewer #1: Yes

Reviewer #2: Yes

5. Review Comments to the Author

Reviewer #1: Economic Burden of Chronic Pain in Alberta, Canada

The manuscript “Economic Burden of Chronic Pain in Alberta, Canada” by Thanh et al. estimated the incremental costs of chronic pain (CP) in Alberta, Canada from a societal perspective. The incremental costs, hence, included healthcare costs and productivity losses. The number of persons with CP was estimated from Canadian Community Health Survey (CCHS) and the incremental healthcare costs per person with CP (compared to non-CP) were estimated from Alberta administrative health data. The productivity loss was estimated from modelling, using data from the CCHS and published data from literature.

Overall, the manuscript is well structured, nicely written, and easy to follow. The methods for estimating the costs (both healthcare and productivity loss) are robust. The study implication is important as its findings will serve the Alberta Pain Strategy 2019-2024.

I have a couple of comments and questions as below:

1. By definition, CP is long standing pain that persists beyond usual recovery period (3 months) as in the introduction. However, the questions of pain and discomfort in the CCHS did not specify time period of pain/discomfort. Therefore, it could be the case that the prevalence using CCHS would be overestimated. Although, in the discussion, the authors cited another study which reported similar prevalence (20.6%), the authors should discuss potential bias of using the CCHS prevalence where a strict definition of CP was not used.

2. Several components of healthcare costs (inpatient, outpatient, physician claims, diagnostic imaging, and drug costs) were estimated from administrative health data though 11 proxy CP-related conditions. However, it’s not clear how these conditions were included. The authors should provide more details on

a. ICD codes for these conditions, probably in Supplementary documents

b. How these conditions (primary diagnosis vs. secondary diagnoses) in inpatient, outpatient, and physicians claims database were included for cost calculation?

c. How the authors identified what diagnostic imaging was for CP patients? Is there a timeframe between an inpatient stay/ outpatient visit/ physician claim with a diagnosis of CP-related conditions and an imaging event to determine that the imaging event was actually for the CP-related conditions?

d. What drugs were included for CP-related drug costs?

Reviewer #2: I would like to thank authors for writing an interesting article. In general, this is a well written and analyzed study. Chronic pain (CP) is highly prevalent, so it is important study topic. However, it is a difficult one, since there is very few accurate methods to estimate costs related to CP. In this study, the authors used methods that were mainly developed and used before, so from the methodological point of view there was not much new in this study. If there are, I would suggest authors to address them more clearly.

Some comments and suggestions:

1) Abstract; could you add to the methods the time interval that you used in your results section. You only once say it is one year. Timeframe is important once we interpret the results.

2) Introduction; It would be good to add one or two sentences about Alberta and its population and healthcare, since Alberta is not very well known internationally.

3) Also you may say shortly what kind of objectives / interventions the provincial pain strategy is planning to implement. At the current text, there is not much link between the economic outcome and the potential interventions. Are they aiming more for prevalence of CP by reducing the severity/intensity of the pain, or is it trying to reduce chronic diseases where CP is associated (as used in the study methods).

4) The study uses CCHS survey data that is pretty comprehensive and representative for the Alberta population. However, it is self reported. At least for some key diseases there are Canadian estimates for prevalence and incidence. I would have liked to see more comparison of your estimates to existing literature. Related to this, you should discuss more in detail, how accurate the survey question for the used 12 CD from Australian study are, since the question asks "have your physician even told you have X disease". So it is not really close estimate from prevalence or incidence.

5) In the results you have several small tables that are (almost) fully reported also in the text. You could easily reduce the number of tables in the main text either just showing some results in text only or combining some of them. Maybe you then could bring one table from suppl material to main text.

6) You mention that the labor variable was from Newfoundland Labrador (NL). I found that rather strange since the NL and Alberta are pretty different considering the labor force (age, unemployment, education, occupation). I suggest that you discuss the validity of this data use in the discussion section.

7) You used the Australian method to estimate CP in CDs. I was not sure if this method is really accurate enough. Also related to point 3 above, this method only finds change if the prevalence is changing, unless you can change the ratios. Could you discuss this considering the measurement after the pain program cost estimation.

6. PLOS authors have the option to publish the peer review history of their article (what does this mean?). If published, this will include your full peer review and any attached files.

Reviewer #1: **Yes: **Dat Tran

Reviewer #2: No

---

## [Author Response · Author response to Decision Letter 0]

21 Jun 2022

please find the attached response to reviewers

---

## [Decision Letter · Decision Letter 1]

25 Jul 2022

Economic Burden of Chronic Pain in Alberta, Canada

PONE-D-22-11788R1

Dear Dr. Thanh,

We’re pleased to inform you that your manuscript has been judged scientifically suitable for publication and will be formally accepted for publication once it meets all outstanding technical requirements.

Kind regards,

Vijay S. Gc, PhD

Academic Editor

PLOS ONE

Additional Editor Comments (optional):

Reviewers' comments:

Reviewer's Responses to Questions

**Comments to the Author**

1. If the authors have adequately addressed your comments raised in a previous round of review and you feel that this manuscript is now acceptable for publication, you may indicate that here to bypass the “Comments to the Author” section, enter your conflict of interest statement in the “Confidential to Editor” section, and submit your "Accept" recommendation.

Reviewer #1: All comments have been addressed

Reviewer #2: All comments have been addressed

2. Is the manuscript technically sound, and do the data support the conclusions?

Reviewer #1: Yes

Reviewer #2: Yes

3. Has the statistical analysis been performed appropriately and rigorously? 

Reviewer #1: Yes

Reviewer #2: Yes

4. Have the authors made all data underlying the findings in their manuscript fully available?

Reviewer #1: Yes

Reviewer #2: Yes

5. Is the manuscript presented in an intelligible fashion and written in standard English?

Reviewer #1: Yes

Reviewer #2: Yes

6. Review Comments to the Author

Reviewer #1: The manuscript is sound and acceptable for publication. All questions and comments in the previous review have been address adequately.

Reviewer #2: I thank authors for their clarifications for my questions. I found them OK and I don't have any further comments for the authors.

7. PLOS authors have the option to publish the peer review history of their article (what does this mean?). If published, this will include your full peer review and any attached files.

Reviewer #1: **Yes: **Dat Tran

Reviewer #2: No

---

## [Editor Report · Acceptance letter]

2 Aug 2022

PONE-D-22-11788R1 

Economic Burden of Chronic Pain in Alberta, Canada 

Dear Dr. Thanh:

I'm pleased to inform you that your manuscript has been deemed suitable for publication in PLOS ONE. Congratulations! Your manuscript is now with our production department. 

Kind regards, 

on behalf of

Dr. Vijay S. Gc 

Academic Editor

PLOS ONE